# Conformational distributions of isolated myosin motor domains encode their mechanochemical properties

Justin R Porter[1], Artur Meller[1], Maxwell I Zimmerman[1], Michael J Greenberg[1], Gregory R Bowman[1,2]*

[1]Department of Biochemistry and Molecular Biophysics, Washington University School of Medicine in St. Louis, St. Louis, United States; [2]Center for the Science and Engineering of Living Systems, Washington University in St. Louis, St. Louis, United States

**Abstract** Myosin motor domains perform an extraordinary diversity of biological functions despite sharing a common mechanochemical cycle. Motors are adapted to their function, in part, by tuning the thermodynamics and kinetics of steps in this cycle. However, it remains unclear how sequence encodes these differences, since biochemically distinct motors often have nearly indistinguishable crystal structures. We hypothesized that sequences produce distinct biochemical phenotypes by modulating the relative probabilities of an ensemble of conformations primed for different functional roles. To test this hypothesis, we modeled the distribution of conformations for 12 myosin motor domains by building Markov state models (MSMs) from an unprecedented two milliseconds of all-atom, explicit-solvent molecular dynamics simulations. Comparing motors reveals shifts in the balance between nucleotide-favorable and nucleotide-unfavorable P-loop conformations that predict experimentally measured duty ratios and ADP release rates better than sequence or individual structures. This result demonstrates the power of an ensemble perspective for interrogating sequence-function relationships.

*For correspondence:
g.bowman@wustl.edu

Competing interests: The authors declare that no competing interests exist.

## Introduction

Myosin motors (*Figure 1A*) perform an extraordinary diversity of biological functions despite sharing a common mechanochemical cycle. For example, myosin-II motors power muscle contraction, whereas myosin-V motors engage in intracellular transport. This diversity is in part due to differences in myosins' tails and light chain-binding domains, which influence properties like localization and multimerization (*Krendel and Mooseker, 2005*). However, some of this diversity is encoded in the motor domains themselves (*Greenberg et al., 2016*). These differences stem from variations in the tunings of the thermodynamics and kinetics of the individual steps of the myosins' conserved mechanochemical cycle, which couples ATP hydrolysis to actin binding and the swing of a lever arm (*De La Cruz and Ostap, 2004*).

Two important and highly variable parameters for motor function are the rate of ADP release, which sets the speed of movement along actin, and the duty ratio, which is the fraction of time a myosin spends attached to actin during one full pass through its mechanochemical cycle. For example, in muscle, myosin-II motors are arranged into multimeric arrays called thick filaments and the individual motors typically have a strong preference for the actin free state (i.e. low duty ratio). These motors quickly detach after pulling on the actin filament to avoid creating drag for other motors in the array, much as a rower quickly removes their oar from the water to minimize drag. In contrast, individual myosin-Va motors have high duty ratios (i.e. prefer the actin-bound state), helping them to processively walk along actin filaments in intracellular transport. Similarly, the speed of myosin

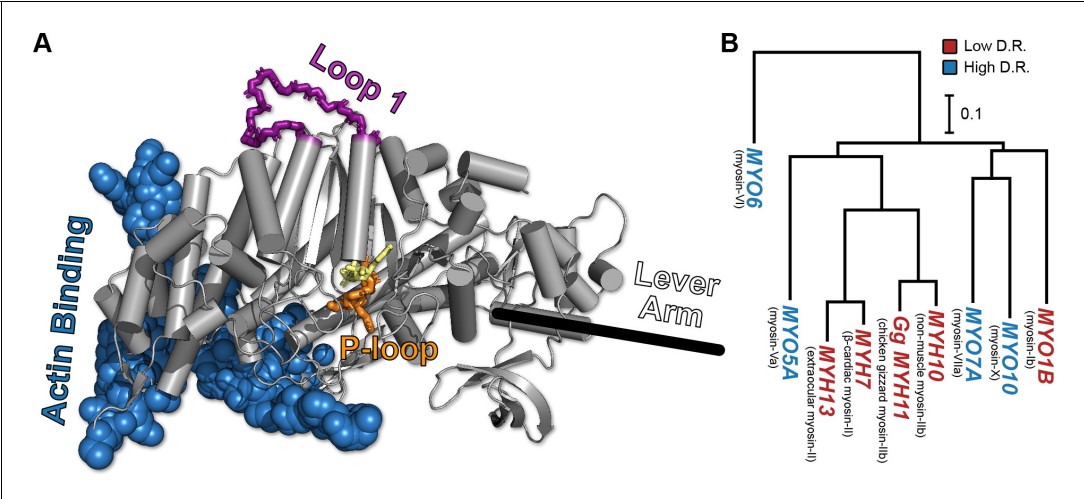

**Figure 1.** The conserved myosin motor domain fold across a diverse phylogeny of motors. (**A**), A crystal structure (PDB ID 4PA0) (*Winkelmann et al., 2015*) of *Homo sapiens* β-cardiac myosin motor domain as an example of the conserved myosin motor domain fold. We note the structural elements most relevant to our work here (loop 1, in purple backbone sticks, and the P-loop, in orange sticks), along with the actin binding region (blue spheres). For orientation, we include the location of the lever arm (black line) and, to indicate the active site, the estimated location of ADP (yellow sticks). (**B**) The phylogenetic relationship the various myosin motor domains examined in this work. Except MYH11, all genes are from *Homo sapiens*. Gene names in blue indicate high duty ratio motors and red indicates low duty ratio. Common protein names are indicated as parentheticals to the left of each gene name. Phylogenetic relationships were inferred from the sequence of the motor domain using *k*-mer distances (*Edgar, 2004a*).

movement along actin (in the absence of opposing forces) is set by the rate of ADP dissociation (*De La Cruz and Ostap, 2004*), and it varies by four orders of magnitude from ~0.4 s$^{-1}$ for non-muscle myosin-IIb (*Nagy et al., 2013*) to >2800 s$^{-1}$ for myosin-XI (*Ito et al., 2007*).

Unfortunately, inferring the relationship between a motor's sequence and its biochemical properties is not trivial. For example, one cannot simply predict the duty ratio or ADP release rate of a motor based on phylogeny. Myosin-V family members contain both high duty ratio motors, like myosin-Va, (*De La Cruz et al., 1999*) and low duty ratio motors, like myosin-Vc (*Takagi et al., 2008*). Similarly, ADP release rates within the myosin-II family vary from ~0.4 s$^{-1}$ (non-muscle myosin-IIb) (*Nagy et al., 2013*) to >400 s$^{-1}$ (extraocular myosin-II) (*Bloemink et al., 2013*; *Johnson et al., 2019*). Insertions and deletions in the myosin motor domain sequence also convey useful, but typically incomplete, information. For instance, pioneering biochemical work *Sweeney et al., 1998* demonstrated a correlation between the length of loop 1 and ADP release rates in myosin-II motors. However, this observation does not explain how other myosin isoforms that have virtually the same loop 1 lengths have ADP release rates that differ by an order of magnitude (*Deacon et al., 2012*). It is also difficult to predict the effects of mutations implicated in human disease, as the effects cannot be easily predicted from the location of the mutation. For example, in human β-cardiac myosin, an A223T mutation causes a dilated cardiomyopathy (*Ujfalusi et al., 2018*) while an I263T mutation has the opposite effect, resulting in a hypertrophic cardiomyopathy (*Tesson et al., 1998*), despite being separated by less than 6 Å (*Planelles-Herrero et al., 2017*).

Structural studies have provided detailed pictures of many key states in the mechanochemical cycle, but have yet to enable the routine prediction of a motor's biochemical properties from its sequence. For example, high-resolution structures have illuminated many shared features of myosin motor domains, such as the lever arm swing (*Fischer et al., 2005*) and conformational rearrangements associated with changes in nucleotide binding (*Coureux et al., 2004*; *Rayment et al., 1993*). They have also revealed the strain-sensing elements of myosin-I motors (*Greenberg et al., 2015*; *Mentes et al., 2018*; *Shuman et al., 2014*) and the binding modes of many small molecules (*Allingham et al., 2005*; *Planelles-Herrero et al., 2017*; *Winkelmann et al., 2015*). However, the structures of motor domains with vastly different biochemical properties are often nearly indistinguishable. Similarly, computer simulations have begun to reveal aspects of motor function (*Blanc et al., 2018*; *Chinthalapudi et al., 2017*; *Hashem et al., 2017*; *Powers et al., 2019*). However, simulating an individual motor domain (~700 residues) is a huge computational expense, so

most simulation studies have been based on less than a microsecond of data. Thus, adding binding partners like actin to simulate the full mechanochemical cycle and infer properties like duty ratio is currently infeasible, especially if one wanted to compare multiple isoforms to infer sequence-function relationships.

Here, we investigate the possibility that the distribution of structures that an isolated motor domain explores correlates with its biochemical properties, allowing the prediction of sequence-function relationships. This hypothesis was inspired by a growing body of work showing that protein dynamics encode function (*Henzler-Wildman and Kern, 2007*; *Knoverek et al., 2019*), even in the absence of relevant binding partners (*Bowman and Geissler, 2012*; *Hart et al., 2016*; *Porter et al., 2019a*). In the case of myosin, we reasoned that as sequence changes modulate motors' preferences for different states of the mechanochemical cycle, they likely also have a systematic effect on the distribution of conformations explored by the motor, even in the absence of binding partners. Therefore, comparing the distribution of conformations that isolated motor domains sample in solution should reveal signatures of their biochemical differences.

To test this hypothesis, we ran an unprecedented two milliseconds of all-atom, explicit solvent molecular dynamics (MD) simulations of twelve myosin motors with diverse but well-established biochemical properties (*Figure 1B*, *Tables 1*, *2*). Such simulations are adept at identifying excited states, which are lower probability conformational states that are often invisible to other structural techniques. Indeed, our simulations reveal a surprising degree of conformational heterogeneity, particularly in the highly conserved P-loop (or Walker A motif), a common structural element for nucleotide binding that is highly conserved across myosin motor domains (*Saraste et al., 1990*). Because of its high conservation, we reasoned that the P-loop would report on the conformation of the nucleotide binding site while still being comparable between motors with otherwise differing sequences. To enable quantitative comparisons, we constructed Markov state models (MSMs) from the MD data for each motor. MSMs are network models of protein free energy landscapes composed of many conformational states and the probabilities of transitioning between these states. They are a powerful means to capture phenomena far beyond the reach of any individual simulation by integrating information from many independent trajectories (*Bowman et al., 2013*; *Chodera and Noé, 2014*). Analyzing our MSMs, we find they capture sufficient information about myosin motor domains' thermodynamics and kinetics to produce reasonable estimates of duty ratio and ADP release rates. Thus, MD and MSMs constitute a powerful platform for identifying relationships between the sequence of individual motor domains and their mechanochemical cycles.

**Table 1.** Summary of simulations performed for this study.

Gene names are those found in PubMed Gene for the appropriate organism, and residue numbers are those used in the given template.

| Gene | Protein name | Construct | Species | Template | Agg. sim [μs] |
|---|---|---|---|---|---|
| MYH13 | Extraocular | 4–781 | *H. sapiens* | 4PA0 (*Winkelmann et al., 2015*) | 271.9 |
| MYH7 | β-cardiac | 2–780 | *H. sapiens* | 4PA0 (*Winkelmann et al., 2015*) | 276.2 |
| MYH10 | Nonmuscle IIb-B2 | 8–791 | *H. sapiens* | 4PD3 (*Münnich et al., 2014*) | 323.0 |
| MYO1B | Myosin-Ib | 5–703 | *H. sapiens* | 4L79 (*Shuman et al., 2014*) | 282.3 |
| MYO5A | Myosin-Va | 2–762 | *H. sapiens* | 1W8J (*Coureux et al., 2004*) | 297.5 |
| MYO6 | Myosin-VI | 2–770 | *H. sapiens* | 2BKI (*Ménétrey et al., 2005*) | 295.0 |
| MYO7A | Myosin-VIIa | 3–742 | *H. sapiens* | 1OE9 (*Coureux et al., 2003*) | 130.9 |
| MYO10 | Myosin-X | 3–740 | *H. sapiens* | 2AKA (*Reubold et al., 2003*) | 126.2 |
| MYH11 | Chicken gizzard | wt/2–782 | *G. gallus* | 4PD3 (*Münnich et al., 2014*) | 6.0 |
| MYH11 | Chicken gizzard | alanine | *G. gallus* | 4PD3 (*Münnich et al., 2014*) | 6.4 |
| MYH11 | Chicken gizzard | *Xenopus* | *G. gallus* | 4PD3 (*Münnich et al., 2014*) | 16.5 |
| MYH11 | Chicken gizzard | Δloop 1 | *G. gallus* | 4PD3 (*Münnich et al., 2014*) | 10.5 |

**Table 2.** Experimentally-determined biochemical properties used in this study.

| Motor | Duty Ratio | ADP Release Rate [s$^{-1}$] | Citation |
|---|---|---|---|
| MYH13 | 0.1 | 400 | *Johnson et al., 2019* |
| MYH7 | 0.1 | 59 | *Johnson et al., 2019* |
| MYH10 | 0.3 | 0.37 | *Nagy et al., 2013* |
| MYO1B | 0.05 | 2.1 | *Lewis et al., 2012* |
| MYO5A | 0.7 | 12 | *De La Cruz et al., 1999* |
| MYO6 | 0.9 | 5.6 | *De La Cruz et al., 2001* |
| MYO7A | 0.9 | 2.1 | *Watanabe et al., 2006* |
| MYO10 | 0.6 | 18 | *Kovács et al., 2005* |
| MYH11, wild-type | 0.15 | 79 | *Sweeney et al., 1998* |
| MYH11, alanine sub. | 0.15 | 34 | *Sweeney et al., 1998* |
| MYH11, *Xenopus* | 0.15 | 40 | *Sweeney et al., 1998* |
| MYH11, Δloop 1 | 0.15 | 13 | *Sweeney et al., 1998* |

## Results and discussion

### In simulation, the P-loop adopts conformational states that are rare in crystal structures

We reasoned that any differences between myosin motor domains in nucleotide handling—ADP release rate or duty ratio, for instance—must somehow be manifest at the active site to have an effect. The P-loop is a highly conserved element of the myosin active site that plays an important role in interacting with the phosphates of the ATP substrate (*Gulick et al., 1997*). Consequently, we reasoned that the P-loop would report on the conformation of the nucleotide binding site while still being comparable between motors whose sequences differ elsewhere in the protein. To assess the degree of conformational heterogeneity captured by crystal structures, we first analyzed structures deposited in the PDB (*Figure 2A*). We queried the PDB (*Berman et al., 2000*) for myosin motor domains (see Materials and methods), yielding 114 crystal structures. Using sequence alignments (see Materials and methods) we identified the P-loop in each of these models and computed the backbone root mean square deviation (RMSD) of each of these models to a reference structure (β-cardiac myosin, PDB ID 4PA0) (*Winkelmann et al., 2015*). We found very little structural diversity among crystal structures, which rarely sample any conformations with P-loop backbone RMSD >0.6 Å away (*Figure 2A*).

Then, to assess the capacity of the P-loop to adopt conformations not observed in crystal structures, we used molecular dynamics to simulate the myosin motor domain. These simulations of human β-cardiac myosin (*Hs MYH7*) were performed in the actin-free, nucleotide-free state for roughly a quarter-millisecond in all-atom explicit-solvent detail used to construct an MSM (see *Methods*). All simulations were conducted using the same force fields and conditions that we have previously used to analyze other systems' conformational distributions, including β-lactamases (*Bowman et al., 2015*; *Porter et al., 2019a*; *Zimmerman et al., 2017*), *E. coli* catabolite activator protein (*Singh and Bowman, 2017*), Ebola virus nucleoprotein (*Su et al., 2018*), and G-proteins (*Sun et al., 2018*). Then, using the MSM, we computed the distribution of backbone RMSDs of the P-loop relative to the reference crystal structure.

In contrast to the relative uniformity among crystal structures, simulations revealed extensive conformational heterogeneity in the P-loop (*Figure 2B*). Where crystal structures rarely sampled conformations with RMSD >0.6 Å, in simulation we observe broad sampling (i.e. high-probability density) in regions from 0.2 Å RMSD all the way to ~1.5 Å RMSD from the starting structure. Only 10 of 114 (9%) crystal structures' conformations were >0.6 Å RMSD from the reference conformation, whereas fully 58% of the distribution observed in silico is above 0.6 Å RMSD from the reference conformation. These results suggest our simulations may provide mechanistic insight not previously accessible from crystal structures alone.

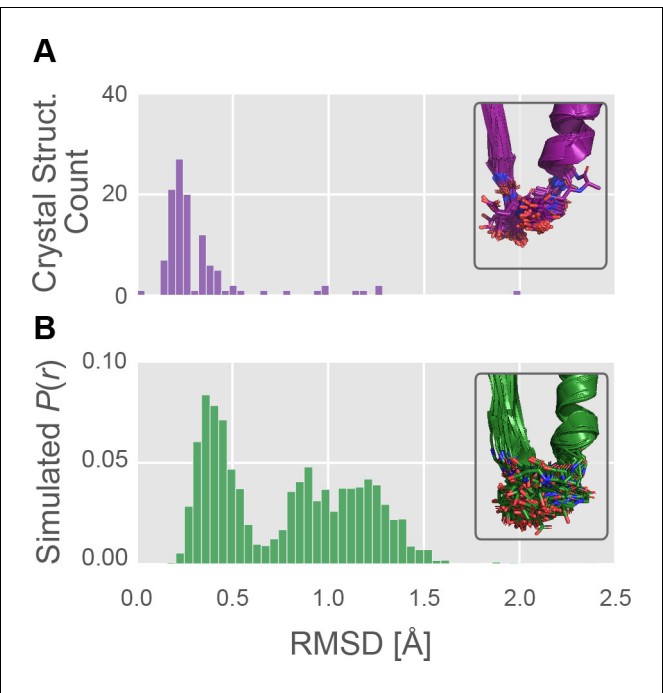

**Figure 2.** The P-loop conformational distribution observed in silico is substantially broader than the distribution found among crystal structures. (**A**) The number of crystal structures (*y*-axis) as a function of P-loop RMSD to 4PA0 (*x*-axis). P-loop conformations in the PDB are largely restricted to backbone RMSD ≤0.6 Å to a reference conformation (PDB ID 4PA0). *Inset*, the 114 myosin crystal structures superimposed, with the P-loop shown as sticks. (**B**) The MSM-derived equilibrium probability (*y*-axis) as a function of P-loop RMSD to 4PA0 (*x*-axis). P-loop conformations from simulations of *Hs* β-cardiac myosin frequently explore conformations that are rare or not seen in crystal structures. *Inset*, the 114 most probable P-loop conformations extracted from our simulations of *Hs* β-cardiac myosin.

## Simulations suggest that the nucleotide-free motor explores distinct nucleotide-favorable and nucleotide-unfavorable states

We reasoned that P-loop conformations identified by our simulations might have important implications for motors' nucleotide handling. For example, modulating the relative probabilities of these conformations would provide a facile mechanism by which sequence variation might tune the mechanochemical cycle.

To assess the nucleotide compatibility of the P-loop conformations we observe in simulation, we sought to systematically compare these conformations with crystal structures with and without nucleotide. To do this, we built a map of P-loop conformational space using the dimensionality reduction algorithm Principal Components Analysis (PCA) to learn a low-dimensional representation of the pairwise interatomic distances between P-loop atoms that retains as much of the geometric diversity in the input as possible (see *Figure 3—figure supplements 1–3*, and Materials and methods for details) (*Shlens, 2014*). We then projected the states of our MSM built from our MYH7 simulations onto principal components (PCs) one and three to visualize the free energy surface sampled by our simulations (*Figure 3A*, green level sets). Principal component two chiefly reported on geometric differences between low-probability confirmations (*Figure 3—figure supplement 1*). Using the same PCA, we then projected each crystal structure's P-loop conformation into the PC1/PC3 space, plotting each as a point (*Figure 3A*, points). Points labeled with PDB IDs represent crystal structures with P-loops >0.6 Å backbone RMSD away from the reference structure 4PA0 used above. We also classified each structure (see Materials and methods) as nucleotide-bound (yellow points) or nucleotide-free (purple points). Then, we compared the frequency at which nucleotide-bound and nucleotide-free P-loop conformations were found in various conformations.

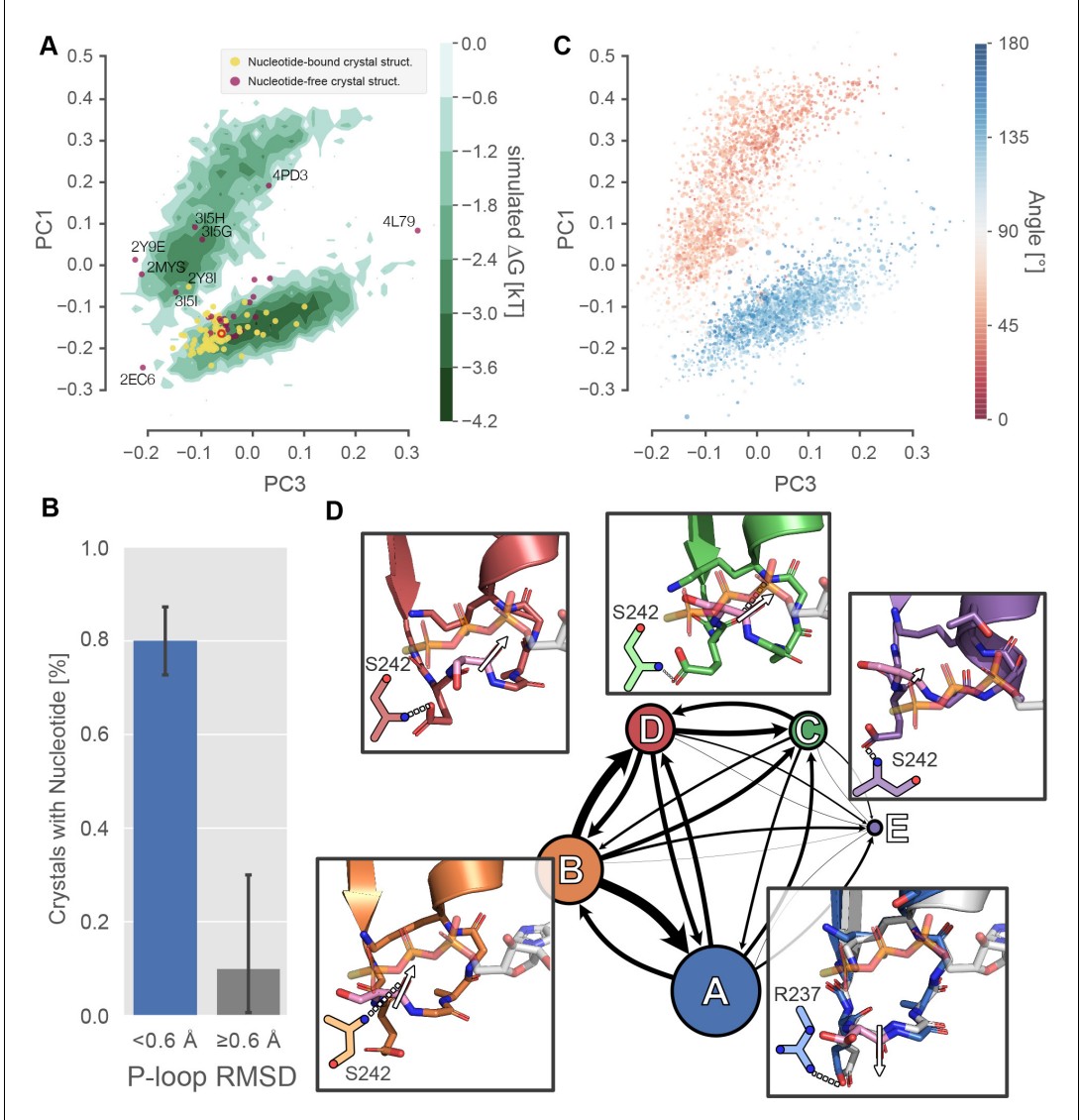

**Figure 3.** Excited P-loop states are less compatible with nucleotide than the states preferred in crystal structures. (A) The β-cardiac whole-motor MSM-derived P-loop conformational space projected onto PCs 1 and 3 reveals two distinct free energy basins (green level sets). Yellow and purple points represent crystal structures with and without ligand, respectively. Structures farther than 0.6 Å from the β-cardiac myosin structure (red empty circle) are labeled with their PDB ID. (B) Proximity to the β-cardiac myosin reference conformation is associated with the presence of a nucleotide in crystal structures (p<$1.3 \times 10^{-5}$ by Fisher's exact test), suggesting that the ligand stabilizes the A state. Error bars represent the 95% confidence interval of 1000 bootstrap realizations. (C) The re-orientation of the S180 backbone carbonyl accounts for the split between upper and lower basins. Points represent P-loop conformations from each state in the β-cardiac whole-motor MSM projected onto the same PCs as in panel A. Points are sized by their probability from the MSM and colored by the angle between the backbone carbonyl bond vectors of S180 and K184. (D) Center, each of the five states of the P-loop MSM are indicated as nodes in a network, sized by their equilibrium probability and connected by arrows with line width proportional to the transition probabilities between them. Surrounding the model, insets show example configurations of the P-loop in sticks colored to match the state they represent. State A is associated with a conformation of the S180 (pink sticks) carbonyl bond vector (white arrow) directed away from the nucleotide binding pocket, whereas states B-D are associated with the opposite orientation of the S180 backbone carbonyl bond vector. The A state conformation is the conformation found in most crystal structures. For reference, PDB 1MMA is shown in grey sticks and the crystallographic position of ATP is shown in semi-opaque grey sticks. For all states, important interactions with the Switch-I loop are shown as two-dimensional sketches for visual clarity. An interaction between R237 and E179 is specific to state A, whereas various interactions with S242 are indicative of other states (*Figure 3—figure supplement 2*).

The online version of this article includes the following figure supplement(s) for figure 3:

**Figure supplement 1.** Variance explained by each component of the PCA of the of the P-loop on MYH7.
**Figure supplement 2.** Joint and marginal distributions for all pairs of PCs.
**Figure supplement 3.** Weights of the first four principal components of the P-loop on MYH7.
*Figure 3 continued on next page*

**Figure supplement 4.** States of the whole-motor MSM of β-lactamase projected into PC1/PC2 and PC1/PC3 planes.
**Figure supplement 5.** Specific interactions with Switch-I residues are statistical hallmarks of P-loop states.

This analysis revealed two dominant conformational states that likely constitute nucleotide-favorable and nucleotide-unfavorable states (*Figure 3A and B*). Once the distribution of P-loop conformations is projected onto two PCs (the green level sets in *Figure 3A*), we observe two broad minima in the P-loop conformational landscape. We refer to these apparent minima as the upper and lower basin for brevity but recognize that other minima may exist and be obscured by the projection of a high-dimensional space into a low-dimensional space. The lower basin (<0.6 Å RMSD from the reference structure) contains 91% of crystal structures (104/114) and, because 80% (84/105) of these structures are bound to nucleotide, it is highly likely to represent a nucleotide-compatible conformation. In contrast, despite being populated roughly equally in simulation, regions outside the lower basin ($\geq$0.6 Å RMSD) contain only 9% (10/114) of crystal structures. And, because only one (11%) of these structures is nucleotide bound, these regions are significantly depleted in nucleotide-bound structures (odds ratio = 0.03, $p < 1.3 \times 10^{-5}$ by Fisher's exact test), strongly implying that they are less or not at all nucleotide compatible. Interestingly, this single exception (PDB ID 2Y8I, *Dictyostelium discoideum* myosin-II G680V) is a highly perturbed motor that has been shown to have low ATPase activity, low motility and a disordered allosteric network (*Kinose et al., 1996*; *Patterson et al., 1997*, p.), potentially contributing to its aberrant conformation.

To characterize the structural differences between nucleotide-favorable and nucleotide-unfavorable states captured in the simulations, we coarse-grained our MSM into a model with just five states, called A-E. We used hierarchical clustering to group the thousands of states explored by *Hs* MYH7 into five states based only on their P-loop conformations (see Materials and methods). Then, using the assignment of each frame from our simulations to one of these five states, we fit a five-state MSM (*Figure 3D*, node sizes indicate equilibrium probabilities, arrow weights indicate transition probabilities). The most probable single state is the A state (49%), which encompasses the entire lower basin and, as we will see below, appears to form favorable interactions with nucleotide based on the conformation of the P-loop. The excited, apparently nucleotide-disfavoring conformations in the upper basin are split into 3 states, B-D, which together account for 50% of the equilibrium probability. Thus, β-cardiac myosin spends about equal time in nucleotide-favorable (state A) and nucleotide-unfavorable states (states B-D) in simulations. Finally, state E (1%, too low to be seen clearly in *Figure 3A*), involves a condensation of the P-loop into an extension of the HF helix, similar to the crystal structure 4L79 (*Shuman et al., 2014*). The reduced number of states in this MSM allowed us to inspect a small number of high-probability conformations near the mean of each P-loop state, which we took as exemplars of each of the five P-loop states.

Comparing the states of our MSM reveals that the dominant geometrical difference between nucleotide-favorable and nucleotide-unfavorable P-loop states is the orientation of the peptide bond between S180 and G181 (*Figure 3C*). In the nucleotide-favorable state A (*Figure 3D*, lower right inset), the S180 backbone carbonyl (shown in pink sticks with a white arrow) is oriented away from the phosphates of the nucleotide, enabling the nucleotide to bind to the active site. In contrast, nucleotide-disfavoring states (labeled B-D in *Figure 3D*) orient the S180 backbone carbonyl toward the phosphate groups of the nucleotide. This positions the carbonyl oxygen in a way that appears to sterically clash with the phosphates of nucleotide. It also orients the negative end of the carbonyl bond's electric dipole toward the nucleotide binding site and the negatively charged phosphates of ADP and ATP. Taken together, our observations about the geometry of the excited, nucleotide-disfavoring state in the upper basin are consistent with a lowered capacity for nucleotide binding.

## The balance between nucleotide-favorable and nucleotide-unfavorable P-loop states predicts duty ratio

We reasoned that motors with a higher probability of adopting nucleotide-favorable P-loop conformations in isolation are likely to have an increased affinity for nucleotide and, therefore, spend more time in nucleotide-bound states of the mechanochemical cycle. Our reasoning is that motors that prefer nucleotide-favorable P-loop conformations in isolation pay a lower energetic cost to adopting

these same nucleotide-favorable conformations when they form a complex with nucleotide. Supporting this logic, it has been observed that, absent load, a large free energy difference between ADP-bound and nucleotide-free states is associated with a low duty ratio (*Bloemink and Geeves, 2011*; *Nyitrai and Geeves, 2004*). Thus, we hypothesized that a preference for the nucleotide-favorable A state should correlate with low duty ratio.

To test if differences in the probability of excited states encodes information about duty ratio, we simulated an additional seven myosin isoforms of differing duty ratio for a total of ~2 ms of aggregate simulation in all-atom, explicit solvent detail. Specifically, we simulated four human low duty ratio myosin motor domains (from myosin-II genes MYH13, MYH7, MYH10, and myosin-I gene MYO1B) and four human high duty ratio myosin motor domains (from genes MYO5A, MYO6, MYO7A, and MYO10), for between 125 and 325 µs each (see Materials and methods). These motors were selected because extensive kinetic characterization (*Bloemink et al., 2013*; *De La Cruz et al., 2001*; *De La Cruz et al., 1999*; *Deacon et al., 2012*; *Homma and Ikebe, 2005*; *Lewis et al., 2012*; *Nagy et al., 2013*; *Watanabe et al., 2006*) has revealed very diverse kinetic tuning, providing a robust test of our hypotheses. Because no crystal structure of the human sequence was available for any of these proteins except MYH7, homology models were built in each case and used as starting points for simulations (see Materials and methods and *Table 1*). To allow for direct comparisons between motors, we used the same PCA and state definitions as described above for MYH7.

As expected, high duty ratio motors have a stronger in silico preference for nucleotide-favoring P-loop states than low duty ratio motors (*Figure 4A*). *Figure 4A* shows an example of this effect on the P-loop conformational distributions of high duty ratio motor MYO6 and low duty ratio motor MYH7. The low duty ratio motor explores both upper and lower basins (*Figure 4A*, left) while the high duty ratio motor strongly prefers the lower basin (*Figure 4A*, right). Provocatively, when motors are crystallized without ligand, only motors with low unloaded duty ratios have been crystallized with P-loops outside the nucleotide-favorable conformation (*Figure 4A*, red and blue points). Of 29 unliganded crystal structures, 8/20 (40%) of low duty ratio motors' P-loops crystallized outside the A state, whereas 0/9 (0%) high duty ratio motors' P-loops crystallized outside state A ($p<0.034$ by Fisher's exact test, see Materials and methods).

Given this trend, we reasoned that the relative free energies of the nucleotide-favorable state and the nucleotide-disfavoring excited states would provide a useful predictor of a motor's duty ratio. We assigned every whole-motor MSM state to one of the five P-loop states and used these assignments to compute the free energies of each of the five states for each of the eight motors (see Materials and methods). We then took the difference in free energy between states A and B, which are the two best sampled states and therefore give statistically robust results. Numerical values and references for these experimental values can be found in *Table 2*.

As expected, we find a strong correlation between motors' duty ratios and their preferences for the nucleotide-favorable A state over the nucleotide-unfavorable B state (*Figure 4B*). Specifically, high duty ratio motors have a strong preference for the A state (negative free energy difference) while low duty ratio motors spend more time in state B (positive free energy difference). Decreased stability of the nucleotide-favorable conformation in these low duty ratio motors could explain this observation.

## Simulations predict ADP release rates better than loop 1 length does by capturing sequence-specific effects

Because ADP release allows a motor to adopt nucleotide-incompatible P-loop conformations, we reasoned that the rate at which a motor can transition to these conformations in silico might correlate with in vitro ADP release kinetics. While we expect a correlation, we acknowledge that the absolute rates will almost certainly differ, since the rates themselves likely differ in the presence and absence of nucleotide. To test for a correlation, we first focus on data sets that examine several motors under the same experimental conditions. Identical conditions are important because in vitro biochemical rates depend strongly on experimental conditions such as salt and temperature (*Chizhov et al., 2013*; *De La Cruz and Ostap, 2009*; *Lewis et al., 2012*). We focus on low duty ratio motors, since their frequent transitions to nucleotide-unfavorable states make it possible to estimate their transition rates with confidence. In contrast, in high duty ratio motors, transitions between these states are sufficiently rare that their rates cannot be estimated with confidence.

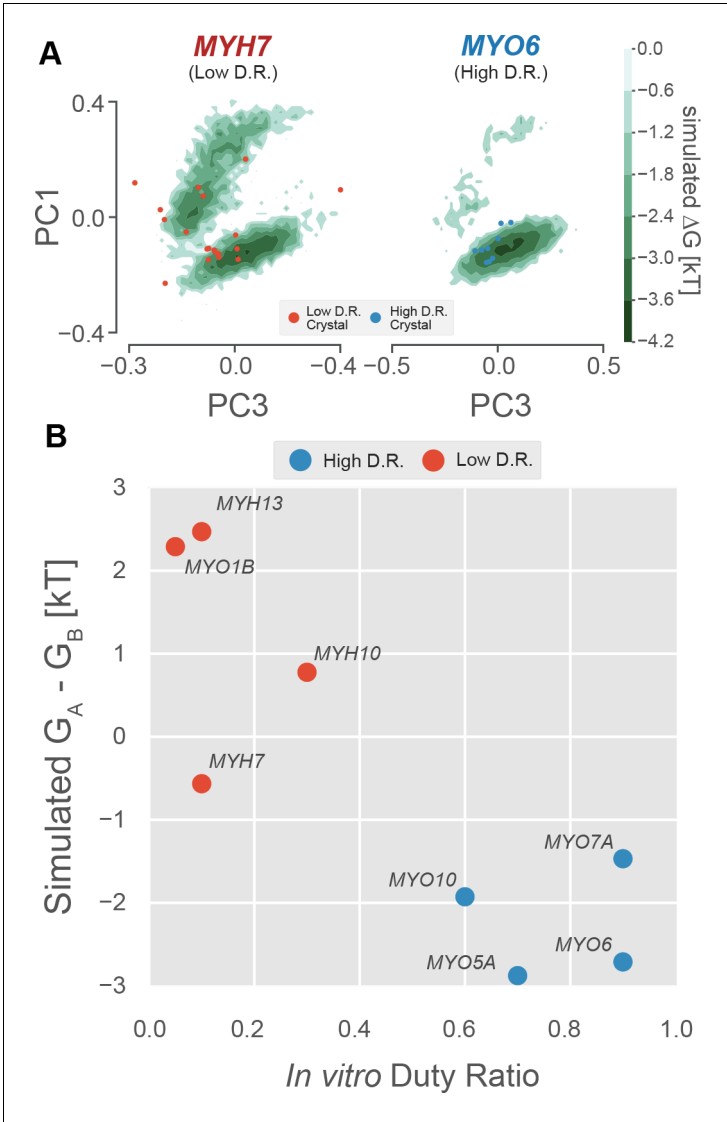

**Figure 4.** The free energy landscape of the P-loop encodes duty ratio. (**A**) Free energy landscapes implied by whole-motor domain MSMs in the PC1/PC3 plane demonstrate that the upper basin is well sampled by an example low duty ratio motor (MYH7, left) and poorly sampled by an example high duty ratio motor (MYO6, right). Ligand-free crystal P-loop conformations from high and low duty ratio motors are shown as blue and red points, respectively. (**B**) Experimental duty ratio (x-axis) is correlated with the simulated free energy difference between (P-loop MSM-derived) nucleotide favorable and nucleotide-unfavorable states (y-axis, more negative values mean higher probability of the nucleotide-favorable A state). Free energy difference is used rather than probability because the log-scaling improves the legibility of the figure. Error in simulated free energy differences were estimated by jackknife resampling of trajectories and were too small to be visualized as error bars. Aggregate simulation times are listed in *Table 3* and were between 126 μs and 323 μs for each point.

An especially useful dataset for comparing relative ADP release rates was created by *Sweeney et al., 1998*, which carefully dissected the effect of variation in loop 1 length and sequence on ADP release rates using the same experimental conditions. These authors established a positive relationship between loop 1 length and ADP release rate using engineered constructs of chicken gizzard myosin-II (shown in *Figure 5A*, henceforth *Gg* MYH11). A notable exception, however, was the myosin with wild-type loop 1, which had an ADP release rate more than three times faster than predicted by the length-based model (*Figure 5B*). This deviation from a purely length-driven ADP release rate led these authors to hypothesize that there must also be sequence-specific effects of

**Table 3.** Parameters of whole-motor Markov state models used in this study.

Fitting coarse-grained P-loop MSMs used the same procedure, but assignments based on P-loop state were used, rather than assignments to whole-motor SASA states. P(A → B) is a parameter of these MSMs. In all cases for coarse-grained P-loop MSMs, a lag time of 37.5 ns was used. Clustering and Markov state model routines are implemented in enspara, git revision f874ba. Solvent accessibility, atomic distance, and RMSD calculations were performed with MDTraj (*McGibbon et al., 2015*). We made extensive use of jug (*Coelho, 2017*) and GNU Parallel (*Tange, 2011*) for task-level parallelization and management of dependencies between tasks.

| Simulation set | No. of states | Cluster radius [nm$^2$] | Lag time [ns] |
| --- | --- | --- | --- |
| MYH13 | 14102 | 7.4 | 0.4 |
| MYH7 | 5128 | 7.34 | 0.5 |
| MYH10 | 7746 | 8.0 | 1.5 |
| MYO1B | 6458 | 6.6 | 0.8 |
| MYO5A | 4728 | 7.25 | 0.4 |
| MYO6 | 4193 | 6.9 | 0.9 |
| MYO7A | 8737 | 6.9 | 0.4 |
| MYO10 | 9273 | 6.9 | 0.4 |
| MYH11, wild-type | 8050 | 4.9 | 1.5 |
| MYH11, alanine sub. | 7822 | 4.9 | 1.5 |
| MYH11, *Xenopus* | 12804 | 5.2 | 1.5 |
| MYH11, Δloop 1 | 8925 | 5.0 | 1.5 |

loop 1 on ADP release rate. They then identified an alanine mutant that ablated the sequence-specific effects of the wild-type loop (henceforth *Gg* MYH11-ala).

To assess the capacity of in silico P-loop kinetics to capture the experimentally measured ADP release rates in the constructs investigated by Sweeny et al, we simulated and analyzed four *Gg* MYH11 constructs. These constructs are a subset of the variants considered by Sweeny et al. We selected the wild-type loop (*Gg* MYH11-wt) because it was the primary outlier in their length-only model. We selected the alanine mutant (*Gg* MYH11-ala) because it, with just five mutations, shifted the wild-type loop in line with the length-only model proposed by Sweeny et al. Then, we selected the extreme points that were well fit by the loop length-only model: the loop 1 deletion (*Gg* MYH11-Δloop1) and the construct using the loop 1 from *Xenopus* non-muscle myosin (*Gg* MYH11-xeno). We simulated these four constructs for 6–16 μs each beginning from a homology model (see Materials and methods and Table S1) and built whole-motor MSMs which, as before, were used to compute five-state P-loop MSMs. Each P-loop MSM contains a parameter P(A→B) which captures the probability that a conformation in state A transitions to state B within a fixed period of time (known as the lag time of the model). We then compared P(A→B) to ADP release rates measured in vitro for these four constructs.

As expected, there is a strong positive relationship (Pearson's R = 0.99) between the P(A→B) fit by our MSMs and in vitro ADP release rate (*Figure 5C*). This is stronger than the equivalent correlation for the length-based model (Pearson's R = 0.72). Importantly, the rank order of the four isoforms is correct, whereas using a loop 1 length-only model dramatically underestimates the ADP release rate for the wild-type motor. Rank order is used because, as noted above, the timescales of the transition (mean first passage times from state A to B are on order of 5–500 nanoseconds) are not directly comparable to experimentally measured values because nucleotide is absent in the simulations. Together, the fact that the sequence change is small (only five residues differ between wild type and the alanine mutant) and the change is distant (~25 Å) from the P-loop indicate that our model is exquisitely sensitive to sequence, even at sites distant from the active site.

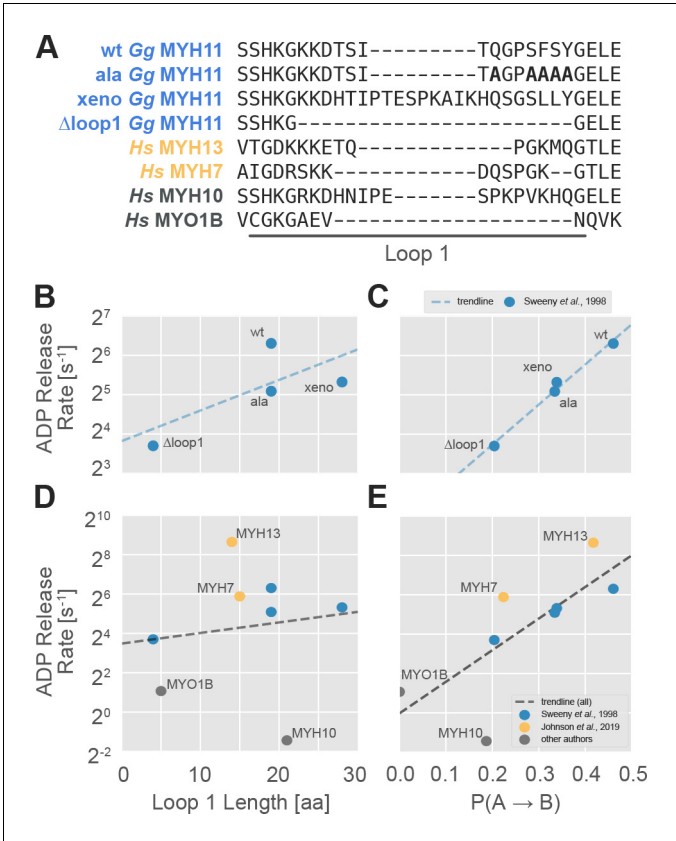

**Figure 5.** The probability of transitioning from nucleotide-favorable to nucleotide-unfavorable P-loop conformations (P(A→B), derived from P-loop MSMs) predicts experimental ADP release rates for motors with low duty ratios. (**A**) Loop 1 sequences and lengths considered in this work. Residues mutated to alanine in the wild-type chicken gizzard MYH11 (wt *Gg* MYH11) are bolded in the appropriate row. (**B**) For the Sweeney dataset, there is a moderate relationship between loop 1 length and ADP release rate (Pearson's R = 0.75) but, (**C**) there is a much stronger correlation between P(A→B) and ADP release rate (Pearson's R = 0.99). (**D**) Across all datasets, the relationship between loop 1 length and ADP release rate is weak (Pearson's R = 0.14), and (**E**) there is a much stronger correlation between P(A→B) and ADP release rate (Pearson's R = 0.75). Error in MSM parameters was estimated by jackknife resampling of trajectories (aggregate simulation times are reported in *Table 3*, and are on the order of ~100 μs for human motors and ~10 μs for chicken motors) and errors in ADP release rates are those reported in the relevant original publication, where available.

## P-loop kinetics in silico correlate with ADP release rates across conditions

To further assess the generalizability of our model, we considered several additional datasets that relax constraints placed on data sets in the previous section. First, we relaxed the constraint that motors differ by just one structural element (loop 1). Specifically, we considered several skeletal myosin isoforms, including MYH7 and MYH13 that Johnson et al (35) studied under the same conditions (*Figure 5D and E*, yellow points). These motor domains are an interesting case because, at 80% sequence identity, their sequences differ much more than Sweeney et al's constructs, and these differences are distributed throughout the protein. Crucially, and despite having roughly the same loop 1 length, their ADP release rates differ by about an order of magnitude (59 s$^{-1}$ vs 400 s$^{-1}$). Owing to the fact that Johnson et al's data were collected under different experimental conditions than Sweeny et al's data (5 mM MgCl$_2$ at 25°C vs 1 mM MgCl$_2$ at 20°C with different light chains), we only expect a general trend to hold, since motors' properties are very sensitive to magnesium, temperature, and light chain identity (*Chizhov et al., 2013*; *Heissler and Sellers, 2014*; *Lewis et al., 2012*). Second, we assessed the trend in two human non-muscle motor domains, MYO1B and

MYH10 with measurements carried out under different conditions. Notably, because they both release ADP very slowly, they test our model's capacity to evaluate very slow ADP release rates.

Consistent with our expectations, and despite the diverse experimental conditions, we still observe a reasonable correlation between P(A→B) and ADP release across all data sets (*Figure 5E*, Pearson's R = 0.75). This dramatically improves on the length-based model (Pearson's R = 0.14). Importantly, under the matched experimental conditions for MYH7 and MYH13 we still find the correct order of ADP release rates (*Figure 3C*, yellow points), suggesting that this method generalizes well to the larger phylogenic distances between myosin isoforms. Furthermore, MYO1B and MYH10 are correctly identified as very slow releasers of ADP, although the point estimates appear to be quite noisy. MYH10 is known to be exquisitely sensitive to light chains (*Heissler and Sellers, 2014*), so it is not surprising that it is one of the greatest outliers given that we did not include these in our simulations.

## Structural models provide insight into the mechanism by which sequence influences P-loop conformational distributions

Even though the sequences of motors' P-loops are identical, their conformational distributions differ. This suggests that interactions with other structural elements in the motor domain bias the P-loop's conformational distribution and our that models capture these effects.

Although no single interaction is likely to completely explain the difference between conformational distributions, to investigate the mechanisms that contribute to this effect we examined the interactions of the P-loop with nearby sidechains. We then compared them between motors to understand how their presence or absence might bias the balance between A and B states for each motor. While an exhaustive analysis is beyond the scope of this work, we have highlighted two examples of such interactions in *Figure 6*.

First, we observed that the A state of the P-loop in the high duty ratio motor MYO6 is stabilized by an interaction between the backbone carbonyl oxygen of the P-loop serine (homologous to S180 in MYH7) and the sidechain amide group of the switch-II residue K670 (*Figure 6A*). A notable difference occurs in the low duty ratio myosin-II motors in our study (MYH13, MYH7, and MYH10), all of which feature an isoleucine at this position (MYH7 I674). Thus, where the strong interaction between the lysine sidechain and P-loop backbone stabilizes the nucleotide-compatible A state in MYO6, this interaction does not exist at all in MYH7, presumably destabilizing this state. *Figure 6B* shows that the sidechain of I674 in MYH7 almost never forms a direct interaction (distance <0.35 nm) with S180 even in P-loop state A, whereas K670 of MYO6 almost always does when the P-loop occupies the A state. We propose that the substitution of an aliphatic residue at this position in myosin-II motors destabilizes the nucleotide-favoring A state, leading to an increased preference for the nucleotide-disfavoring B state, ultimately resulting in a lower predicted duty ratio. Notably, however, many other low duty ratio myosin classes, such as MYO1B which we simulated here and correctly identify as a low duty ratio motor, feature a lysine at this position, implying that this substitution may be a peculiar innovation limited to myosin-IIs.

Second, we also observed that the B state of the P-loop in the low duty ratio motor MYH7 is stabilized by an interaction between the backbone carbonyl oxygen of S242 (in the Switch-I loop) and the S180 sidechain hydroxyl group (*Figure 6C*), but that this interaction does not occur in the high duty ratio motor MYO7A (*Figure 6D*). This interaction is specific to the nucleotide-disfavoring B state of MYH7 (*Figure 6D*), and hence presumably stabilizes that P-loop state in MYH7 relative to MYO7A. As shown in *Figure 6C*, this interaction in MYH7 requires the Switch-I loop to move 'inwards,' toward the peptide bond between G464 and A463, with which it also sometimes interacts. At the position homologous to A463, however, the high duty ratio motor MYO7A features a phenylalanine (F439). We propose that the bulky, aromatic sidechain in MYO7A (F439) prevents the Switch-I loop from engaging the P-loop serine's sidechain, whereas the small aliphatic one (*MYH7*'s A463) does not. On net, this leads to a lower overall preference for the nucleotide-disfavoring B state in MYO7A and thus a higher overall duty ratio prediction. Interestingly, MYO6 has an alanine at this position, indicating that this substitution is not strictly required for high duty ratio.

These examples demonstrate how physically realistic, atomically detailed models can provide mechanistic insight into how sequence variation modulates specific interactions to alter a protein's function. Of course, there are many interactions at play, and consideration of multiple interactions is necessary to fully explain duty ratio. Therefore, a successful pipeline for predicting duty ratio

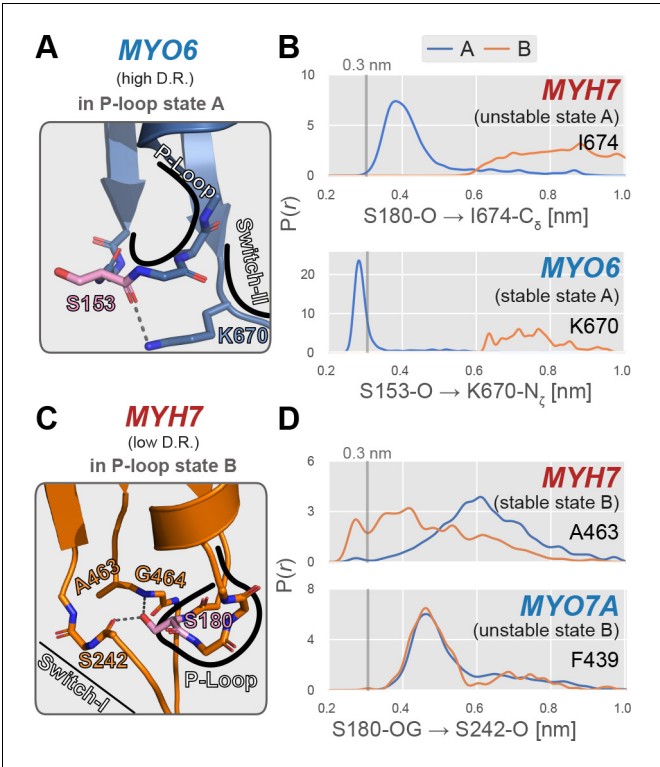

**Figure 6.** Examples of specific P-loop interactions that appear to modulate stability of specific P-loop states. (A) MYO6, a high duty ratio motor, features an interaction between the backbone carbonyl oxygen of S153 (analogous to S180 in MYH7) and the side chain amide group of K670 on the switch-II loop. Myosin-IIs (MYH13, MYH7, and MYH10) feature an isoleucine at this position, eliminating this A state-stabilizing interaction. (B) In MYO6, the sidechain of K670 interacts tightly with the S153 backbone, whereas in MYH7, there is virtually no direct interaction between the homologous positions (S180 and I674). This presumably leads to a stabilization of state A in MYO6, contributing to its preference for the A state over the B state. An approximate cutoff distance for interaction, 3 Å, is marked as a vertical line. (C) In MYH7, a low duty ratio motor with a high relative B state probability permits interaction between the backbone carbonyl oxygen of S242 and the sidechain hydroxyl group of S180. This geometry is permitted by a small alanine sidechain at position 463. (D) When the P-loop is in state B, the sidechain hydroxyl group of S180 hydrogen bonds with the backbone carbonyl oxygen of S242 in MYH7, whereas MYO7A's homologous positions do not interact in either state. This interaction presumably increases the relative probability of the B state relative to the A state and could in part account for MYH7's higher propensity to adopt this excited conformation. The reference distance 3 Å is marked as a vertical line.

predictions will probably require additional molecular dynamics simulations for any new variant. Specifically, we suggest that a fruitful design strategy would be to select mutations based on logic like that outlined above, then to simulate the newly designed sequences to check that they behave as intended (in silico), and then to perform experimental tests of these predictions. Such an approach has proved powerful in past applications to other proteins (*Hart et al., 2016*; *Zimmerman et al., 2017*).

## Conclusions

In this work, we used computer simulations of isolated myosin motor domains to predict the in vitro ADP release rate and duty ratio of unloaded myosin motors. To do this, we identified systematic shifts in the distribution of conformations that a motor explores that correlate with changes in biochemistry, rather than by directly simulating the biochemical processes themselves, which would have been prohibitively expensive. While binding partners (actin and nucleotide, for instance) and structural elements outside the motor domain almost certainly affect the distribution of conformations, our results demonstrate that it is nevertheless possible to extract reasonable estimates for at least some unloaded biochemical properties from only the isolated motor domain's conformational

distribution. The ability of the isolated motor domain's fluctuations to predict these parameters likely stems from a link between the isolated and bound conformational distributions. In other words, because the motor domain active site must adopt certain key conformations during its functional interactions with binding partners (i.e. nucleotide and actin), it is nearly guaranteed to at least transiently sample those conformations even in the absence of those binding partners. Importantly, our simulations only require a reasonable homology model as a starting point, so our methods should be applicable to a broad range of motor variants, including mutations implicated in disease.

Given the high degree of structural conservation of the myosin motor domain, it was not previously possible to directly predict the duty ratio or kinetics for a given myosin isoform from the sequence or structure of a motor domain alone. Our studies demonstrate that the duty ratio and the rate of ADP release are not captured by a single structural element, but rather by the distribution of conformations that the motor explores in solution. Throughout our simulations, we observed that the distribution of P-loop conformations is sensitive to relevant sequence changes, both large and small, throughout the myosin motor domain. Presumably, these changes are allosterically propagated through the myosin motor domain through complex networks of coupled motions. Thus, capturing the difference between the wild-type and alanine-substituted chicken gizzard myosins (*Figure 5C*), for instance, required the model to capture the allosteric perturbation induced by a change of a few dozen atoms in a molecule of ~12,500 atoms at a distance of ~25 Å (*Figure 1A*). Meanwhile, classifying the duty ratio of diverse myosin motors requires the P-loop to integrate signals from across the molecule into a single overall conformational preference. This underscores a key advantage of physics-based simulations, which is the ability to represent these allosteric networks by modeling in detail the complex, nonlinear couplings throughout the molecule.

One tantalizing interpretation of the excited states of the P-loop we observe in silico is that they may be related to the biochemically-observed 'open' and 'closed' states that nucleotide-free myosin motors populate in vitro (*Geeves et al., 2000*). In our simulations, we see that the P-loop fluctuates between conformations that are nucleotide-compatible and conformations that probably are not. In biochemical experiments, at least some myosin isoforms in the nucleotide-free actin-bound state fluctuate between a state that binds nucleotide and a state that does not. It has also been shown that the equilibrium between these two biochemical states ($K_\alpha$), correlates with duty ratio and the transition rate from the nucleotide binding incompetent state to the nucleotide binding competent state ($k_{+\alpha}$) correlates with the ADP release rate (*Bloemink and Geeves, 2011*). Similarly, we showed that the equilibrium between nucleotide-favorable and nucleotide-disfavorable conformations predicted duty ratio, while the rate of transition predicted ADP release rate. A simple explanation for these similarities is that there may be a correspondence between these biochemical states and the structural states that we observe in our MSMs in silico.

Finally, our results highlight the general capacity of computational modeling to link sequence and function. One immediate application of our work here is to estimate in silico the biochemical parameters of new or difficult-to-study myosins. In the near term, constructing such models could help us learn more about the atomic basis for healthy functional diversity in myosin motors, and how small changes can give rise to malfunction and disease. Indeed, in the coming years it may prove possible to use these models as a tool for studying patient-specific mutations by understanding the atomic basis for diseases caused by dysfunction of myosin motors or to aid in developing therapeutics. Finally, because we find no reason to believe our approach's applicability is limited to myosin motors, we expect the techniques we have presented here to be of use for any protein where the physics that maps sequence to biochemistry is not straightforward.

## Materials and methods

### Preparation of homology models

For simulations, the initial structure of each myosin motor domain was prepared by first obtaining the full-length protein's sequence from PubMed Protein, trimming the sequence down to include only the motor domain using crystal structure 4PA0 of MYH7 as a guide, and submitting that sequence to SWISS-MODEL for homology modeling (*Waterhouse et al., 2018*). Templates were chosen with a preference for those that were high-resolution, high sequence similarity, and in the rigor state. A complete list of sequences, templates, and motor domains can be found in *Table 1*.

### Preparation of example myosin conformation

In *Figure 1A*, the position of ATP is based on ligand-bound crystal structure 1MMA (*Münnich et al., 2014*). The actin binding region was defined by all atoms within 10 Å of the actin filament after alignment to 6BNP chain K (*Gurel et al., 2017*).

### Sequence alignments

All sequence alignments were performed with MUSCLE 3.8.1551 (*Edgar, 2004b*) using default parameters. Phylogenetic trees were inferred with the neighbor joining method using these alignments. Distances between sequences were *k*-mer distances (*Edgar, 2004a*).

### Molecular dynamics simulations

GROMACS (*Abraham et al., 2015*; *Berendsen et al., 1995*) was used to prepare and to simulate all proteins. The protein structure was solvated in a dodecahedron box of TIP3P water (*Jorgensen et al., 1983*) that extended 1 nm beyond the protein in every dimension. Thereafter, sodium and chloride ions were added to produce a neutral system at 0.1 M NaCl.

Each system was minimized using steepest descents until the maximum force on any atom decreased below 1000 kJ/(mol × nm). The system was then equilibrated with all atoms restrained in place at 300°K maintained by Bussi-Parinello thermostat (*Bussi et al., 2007*). After these equilibration runs, the restraints on heavy atoms were removed.

Molecular dynamics were performed using the AMBER03 force field (*Duan et al., 2003*). All covalent bonds involving hydrogen were constrained using LINCS (*Hess et al., 1997*). Virtual sites were used to allow for a 4 fs time (*Feenstra et al., 1999*).

Production simulations were performed on a mixture of Folding@home (*Shirts and Pande, 2000*) and an in-house supercomputing cluster. A mix of Tesla K20, Titan Xp, Tesla P100, and Quandro RTX 6000 GPUs were used and Intel Xeon E5-2650 v2, Intel Xeon E5-2630 v3, Intel Xeon E5-2690 v4, Intel Xeon Gold 6148 CPUs clocked at 2.4–2.6 GHz were used. Using GROMACS 2019.2, nodes featuring a Tesla K20 or Titan Xp produced ~22 ns/day, nodes featuring a Tesla P100 produced ~61 ns/day, and nodes featuring a Quadro RTX 6000 produced ~95 ns/day.

### Markov state models

Fine-grain, whole-motor domain Markov state models were constructed first by defining microstates using the *k*-hybrid clustering algorithm with five rounds of *k*-medoids refinement using the Euclidean distance between residue sidechain solvent accessible surface area (scSASA) as a distance metric. This approach first appeared in *Porter et al., 2019a* and was chosen because it scales well for extremely large datasets compared to traditional RMSD clustering. The reasons for this are discussed in *Porter et al., 2019b* but, briefly, although scSASA calculations are initially expensive, they realize substantial performance gains in clustering because each frame's scSASA need only be computed once. ach frame can be computed independently, allowing for massive parallelization. It also reduces the size of the input data size, since only a single floating point number represents an entire residue, and allows the use of a cheaper distance metric (Euclidean distance rather than RMSD).

Markov state models were then fit for each variant by applying a $1/n$ pseudocount to each element of the transition counts matrix and row-normalizing, as recommended in *Zimmerman et al., 2018*. Lag times were chosen by the implied timescales test and by examining the equilibrium probability distribution for unrealistically overpopulated states (suggesting insufficient sampling of a particular transition or internal energy barriers). Important hyperparameters are listed in *Table 3*.

### Construction of the P-loop free energy surface

Pairwise interatomic distances in the P-loop were computed using MDTraj (*McGibbon et al., 2015*), selecting all possible pairs of a backbone amide nitrogen and a backbone carbonyl oxygen atom in the GESGAG portion of the Walker A motif (*i.e.*, the conserved P-loop sequence) that makes up the P-loop.

Principal components analysis (PCA) was performed on the 36-dimensional pairwise atomic distance vectors for each MSM microstate using the PCA implementation in sklearn (*Pedregosa et al., 2011*). No whitening was employed and the full SVD was calculated.

The surface was then estimated by constructing a weighted two-dimensional histogram in the PC1/PC3 plane with 50 bins between the minimum and the maximum data in each direction. The resulting array of probabilities was then converted into free energies of units $kT$ by taking the natural logarithm of each value. It was then convoluted with a gaussian of variance 0.3 per grid cell using scipy's gaussian_filter method (*Oliphant, 2007*). The resulting array was then level-set into six level sets.

## Selection of myosin motor domain PDB crystal structures

We selected crystal structures to map on to the P-loop free energy landscape by querying the PDB (*Berman et al., 2000*) for all structures with sequence identities to the motor domain of *Hs MYH7* greater than 10%, resolution <= 5.0 Å and a BLAST E-value less than $10^{-10}$. We then selected the largest chain in each crystal structure, used muscle (*Edgar, 2004b*) to align that chain's sequence to the motor domain of *Hs MYH7*, and used the resulting alignment to identify the P-loop. P-loop distances were computed and projected into the low-dimensional space as described above. Sequence bookkeeping and I/O relied heavily on scikit-bio (github.com/biocore/scikit-bio; *scikit-bio development team, 2014*).

Crystal structures were classified as bound to a nucleotide or nucleotide analogue if they contained a residue with the name ADP, ATP, ANP, MNQ, MNT, ONP, PNQ, DAE, DAQ, NMQ, AGS, AD9, AOV, or FLC.

## Hierarchical clustering of the P-loop

The five coarse-grained MSM microstates for MYH7 were learned using agglomerative clustering on the four-dimensional P-loop features learned by PCA for the free energy surface. Ward linkage and a Euclidean distance metric were used. Briefly, the states are recursively combined in a way that minimizes the within-cluster variance in a until the specified number of clusters is reached. The number of clusters were increased until no obvious internal free energy barriers were seen in the four PC dimensions. Agglomerative clustering was implemented by sklearn 0.21.2 (*Pedregosa et al., 2011*).

## Assignment of new conformations to P-loop states

P-loop state assignments for conformations of motors other than *Hs* MYH7 were made using a *k*-nearest neighbors (*Pedregosa et al., 2011*) approach. In this approach, a query conformation is assigned to a cluster based on the assignments of nearest *k* points in the labeled dataset (i.e. MYH7). In other words, the nearest *k* points to the query point 'vote' on the assignment of the query point to a cluster. In our case, *k* was 5, but we did not appreciate any differences for values of *k* from 3 to 15.

Implementation of *k*-nearest neighbors was from sklearn 0.21.2. A ball tree was used to speed the search for neighbors (*Omohundro, 1989*).

## Estimation of equilibrium probability of P-loop states

For each motor, the probability of a P-loop state was calculated by summing the equilibrium probabilities of all states in the whole-motor MSM assigned to that P-loop state.

## Biochemical properties of myosin motors

For each of the human myosin motors we simulated, an experimental duty ratio is available for either human or a vertebrate relative (e.g. cow, chicken) motor. Thus, wherever numerical duty ratios are reported (e.g. *Figure 4B*), these biochemical measurements are used. The experimentally-measured duty ratios and ADP release rates used in this work are shown in *Table 2*.

In our analysis of duty ratio and P-loop crystal position in *Figure 4A*, some constructs' unloaded duty ratios have not been measured. For these motors, it was therefore necessary to infer whether they have high or low duty ratios from phylogeny. Specifically, we plotted: 4DBP, 2MYS, 3I5H, 2Y0R, 2BKH, 6I7D, 1DFK, 1OE9, 3I5I, 2OS8, 4P7H, 5V7X, 4ZLK, 1MNE, 1FMV, 2AKA, 3MYL, 2EC6, 4L79, 3L9I, 2BKI, 2Y9E, 1KK7, 1W8J, 2 × 51, 4PA0, 4PD3, 3I5G, and 1SR6. Based upon previous biochemical experiments, myosin-Is and IIs were assumed to have low duty ratios. Myosin-VIs were assumed to have high duty ratio. Myosin-Va and Vb from all organisms were assumed to have high duty ratios

and Myosin-Vc was assumed to have a low duty ratio. *Plasmodium falciparum* MyoA (6I7D) has been shown to have a high duty ratio (*Robert-Paganin et al., 2019*).

Myosin class was inferred as follows. Where a roman numeral was given in the PDB description (e.g. Myosin-II) this classification was used. Otherwise, if 'muscle' or 'striated' was appeared in the PDB polymerDescription field, the myosin was classified as a myosin-II. Finally, in the absence of other indicators, myosins from *Doryteuthis pealeii*, *Placopecten magellanicus*, and *Argopecten irradians* were classified as Myosin-IIs, and myosins from *Plasmodium falciparum* were classified as Myosin-XIVs.

## Visualization

Proteins structures were visualized and rendered with PyMOL. Data plots were constructed with matplotlib (*Hunter, 2007*). Free energy surface colormaps were constructed with the cubehelix color system (*Green, 2011*).

## Code and model availability

MSMs and starting conformations for each of the myosin constructs studied in this have been uploaded to the Open Science Framework as project ID 54 G7P, along with the parameters for the PCA used in *Figures 2* and *3*. This OSF project also includes a CSV that lists the P-loop definition, P-loop RMSD from the reference state, and assignment to P-loop state A-E for each crystal structure.

## Acknowledgements

We are extremely grateful to the citizen scientists of Folding@home for their generous donation of computing resources. We are also grateful to Prof. Eric Galburt, Prof. John Edwards, and Dr. Joshua Alinger for their insight and helpful comments about this work. We are also grateful to the Center for High Performance Computing at the Mallinkrodt Institute for Radiology for computer time.

This work was funded by National Institutes of Health grants R01GM12400701 (GRB), R01HL141086 (MJG), T32GM02700 (AM), and F30HL146052 (JRP), National Science Foundation CAREER Award MCB-1552471 (GRB), GRB holds a Career Award at the Scientific Interface from the Burroughs Wellcome Fund and a Packard Fellowship for Science and Engineering from the David and Lucile Packard Foundation (GRB). MIZ was supported in part by a Monsanto Graduate Fellowship and a Center for Biological Systems Engineering Fellowship.

## Additional information

### Funding

| Funder | Grant reference number | Author |
| --- | --- | --- |
| National Institutes of Health | R01GM12400701 | Gregory R Bowman |
| National Institutes of Health | R01HL141086 | Michael J Greenberg |
| National Institutes of Health | T32GM02700 | Artur Meller |
| National Institutes of Health | F30HL146052 | Justin R Porter |
| National Science Foundation | MCB-1552471 | Gregory R Bowman |
| Burroughs Wellcome Fund | Career Award at the Scientific Interface | Gregory R Bowman |
| David and Lucile Packard Foundation | Fellowship for Science and Engineering | Gregory R Bowman |
| Monsanto Company | Graduate Fellowship | Maxwell I Zimmerman |
| Washington University in St. Louis | Center for Biological Systems Engineering Fellowship | Maxwell I Zimmerman |

The funders had no role in study design, data collection and interpretation, or the decision to submit the work for publication.

### Author contributions
Justin R Porter, Conceptualization, Resources, Data curation, Software, Formal analysis, Supervision, Funding acquisition, Validation, Investigation, Visualization, Methodology, Writing - original draft, Project administration, Writing - review and editing; Artur Meller, Conceptualization, Data curation, Formal analysis, Visualization, Writing - original draft, Writing - review and editing; Maxwell I Zimmerman, Conceptualization, Software, Supervision, Visualization, Methodology, Writing - review and editing; Michael J Greenberg, Conceptualization, Formal analysis, Supervision, Funding acquisition, Investigation, Methodology, Writing - original draft, Project administration, Writing - review and editing; Gregory R Bowman, Conceptualization, Supervision, Funding acquisition, Methodology, Writing - original draft, Project administration, Writing - review and editing

### Author ORCIDs
Justin R Porter (iD) https://orcid.org/0000-0002-0340-951X
Artur Meller (iD) http://orcid.org/0000-0002-5504-2684
Michael J Greenberg (iD) http://orcid.org/0000-0003-1320-3547
Gregory R Bowman (iD) https://orcid.org/0000-0002-2083-4892

### Decision letter and Author response
Decision letter https://doi.org/10.7554/eLife.55132.sa1
Author response https://doi.org/10.7554/eLife.55132.sa2

## Additional files

### Supplementary files
• Transparent reporting form

### Data availability
MSMs and starting conformations for each of the myosin constructs studied in this have been uploaded to the Open Science Framework as project ID 54G7P, along with the parameters for the PCA used in Figures 2 and 3. This OSF project also includes a CSV that lists the P-loop definition, P-loop RMSD from the reference state, and assignment to P-loop state A-E for each crystal structure.

The following dataset was generated:

| Author(s) | Year | Dataset title | Dataset URL | Database and Identifier |
|---|---|---|---|---|
| Porter JR, Meller A, Zimmerman MI, Greenberg MJ, Bowman GR | 2020 | myosin-isoforms | https://osf.io/54g7p/ | Open Science Framework, 54g7p |

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
