## [Decision Letter]

Thank you for submitting your article "Conformational distributions of isolated myosin motor domains encode their mechanochemical properties" for consideration by *eLife*. Your article has been reviewed by three peer reviewers, and the evaluation has been overseen by José Faraldo-Gómez as the Senior/Reviewing Editor. The reviewers have opted to remain anonymous.

The reviewers have discussed the reviews with one another and the Senior Editor has drafted this decision to help you prepare a revised submission.

Summary:

This work provides important insights into a question of broad significance in molecular biophysics, namely how the functional mechanisms of proteins emerge from their structural dynamics. The authors carry out extensive molecular dynamics simulations of several myosin motor domains to probe the conformational distribution of the nucleotide binding site. The results are used to construct the free energy landscape and kinetics for the transition between different metastable states in the general framework of Markov State models (MSM). The underlying hypothesis is that the intrinsic free energy landscape of the binding site (or the P-loop, in particular) in the absence of nucleotide contributes to explain the nucleotide binding properties of the motor domain. In this perspective, key properties of the motor domain that are correlated with functional features, such as duty ratio, are encoded in sequence. The hypothesis is supported by the observed correlation between experimentally measured (in previous literature) duty ratios and ADP release rates with the computed free energy difference between the two key basins, which correlate with the nucleotide compatible and nucleotide-free structural states. Overall, the study is a good demonstration of how extensive, state-of-the-art MD simulations can be used to capture somewhat subtle yet mechanistically relevant features in complex molecular systems. It is easy to envisage how a similar methodology could be used to probe other biological problems that involve shift of populations among different structural states. The work is also a good illustration of how to leverage information from homology modeling, i.e. through a systematic analysis of general mechanistic features in a family or collection of models, as opposed to detailed aspects specific of given protein, likely to be beyond the accuracy of such models.

Essential revisions:

1) The analysis provided points towards a specific peptide bond whose dynamics causes the backbone carbonyl oxygen of S180 to point either towards the nucleotide binding site or away from it. It would be of great interest to provide an experimentally testable hypothesis by predicting a mutation that specifically determines the conformational preference of this peptide bond, thereby shifting the activity either up or down for a particular myosin.

2) In Figure 4B, the authors claim the error bars are too small to be seen. This is surprising given that the underlying data for each point is a few microseconds of MD simulation. The authors should carefully explain and justify their error analysis. If the MD data is fragmented into, say, 10 parts, do the MSM deduced from each fragment yield indistinguishable free-energy difference values? Showing some trajectories in which the system switches between basins might help make this point also.

3) In the whole-motor Markov models, a lag time of only ~1 ns (Table 2) does not seem immediately intuitive. This choice requires some kind of rationale and, ideally, data to support it.

4) The authors indicate that phylogeny relationships were used to infer duty ratios. This approach needs justification and validation, if indeed it was actually used in the data shown – in which case inferred values should be noted (not the same as a measurement).

5) In their analysis, the authors identify an interesting mutation distant from the active site that nevertheless affects the ADP off rate, for which the transition rate P(A->B) is taken as proxy. Do the simulations provide insight into how this distant mutation alters the barrier for the A->B transition?

6) Given that the identified thermodynamic (duty ratio) and kinetic (ADP off rate) functional readouts point towards the same A-B transition, do the data allow to speculate whether the two are inherently linked to each other or whether they might have evolved independently?

---

## [Author Response]

Essential revisions:1) The analysis provided points towards a specific peptide bond whose dynamics causes the backbone carbonyl oxygen of S180 to point either towards the nucleotide binding site or away from it. It would be of great interest to provide an experimentally testable hypothesis by predicting a mutation that specifically determines the conformational preference of this peptide bond, thereby shifting the activity either up or down for a particular myosin.

To address this question, we have added subsection “Structural models provide insight into the mechanism by which sequence influences P-loop conformational distributions.” to the Results and Discussion, along with an additional figure (Figure 6), which outline the ways that sequence might be driving the differences we observed. In particular, we identify two examples of changes to the sequence of the myosin motor active site that appear to modulate the probability of the A and B states of the P-loop, presumably shifting the balance of these two states for particular motors one way or the other.

2) In Figure 4B, the authors claim the error bars are too small to be seen. This is surprising given that the underlying data for each point is a few microseconds of MD simulation. The authors should carefully explain and justify their error analysis. If the MD data is fragmented into, say, 10 parts, do the MSM deduced from each fragment yield indistinguishable free-energy difference values? Showing some trajectories in which the system switches between basins might help make this point also.

The reviewers are correct—such small error bars would be unlikely for simulations of just a few microseconds. In the case of Figure 4B (the figure in question), each point is based on ~250 µs. We have clarified this point in the text:

“Error in simulated free energy differences were estimated by jackknife resampling of trajectories and were too small to be visualized as error bars. Aggregate simulation times are listed in Table 2 and were between 126 µs and 323 µs for each point.”

3) In the whole-motor Markov models, a lag time of only ~1 ns (Table 2) does not seem immediately intuitive. This choice requires some kind of rationale and, ideally, data to support it.

We agree this is an important issue. We have therefore noted in the Materials and methods section:

“… Lag times were chosen by the implied timescales test…”

This is a standard practice in the field, in which one examines the slowest relaxation rates as a function of the lag time used. If the state definitions are sufficient to satisfy the Markov assumption, then these “implied timescales” will become invariant with respect to the lag time. In our case, this invariance is achieved by lag times of 1 ns and greater.

In addition, much previous work on this type of question has also used lag times on the order of ~1 ns (see, for example, Zimmerman et al., 2017; Singh and Bowman, 2017; Sun et al., 2018; Porter et al., 2019a).

4) The authors indicate that phylogeny relationships were used to infer duty ratios. This approach needs justification and validation, if indeed it was actually used in the data shown – in which case inferred values should be noted (not the same as a measurement).

This is a crucial point and we thank the reviewers for noting this ambiguity. To clarify this, we have added the following to the text:

“For each of the human myosin motors we simulated, an experimental duty ratio is available for either human or a vertebrate relative (e.g. cow, chicken) motor. Thus, wherever numerical duty ratios are reported (e.g. Figure 4B), these biochemical measurements are used.”

5) In their analysis, the authors identify an interesting mutation distant from the active site that nevertheless affects the ADP off rate, for which the transition rate P(A->B) is taken as proxy. Do the simulations provide insight into how this distant mutation alters the barrier for the A->B transition?

We are very interested in this question. Our preliminary results suggest the mechanism is nontrivial, however, and we plan to address the allostery between the active site and loop 1, as well as between these regions and other sites, such as the actin binding interface, converter domain, and relay helix in future work.

6) Given that the identified thermodynamic (duty ratio) and kinetic (ADP off rate) functional readouts point towards the same A-B transition, do the data allow to speculate whether the two are inherently linked to each other or whether they might have evolved independently?

This is a fascinating question! On one hand, because the ADP off rate sets the rate of actin detachment, and duty ratio is the balance between actin attachment and detachment, there is an inextricable link between duty ratio and actin detachment. On the other hand, it is well-known (e.g. De La Cruz and Ostap, 2009) that duty ratio (thermodynamics) and ADP release (kinetics) appear to be under independent selective pressure. In fact, surprisingly, the fastest-releasing known myosin is a high duty ratio myosin.